

**First reported case of Thunderstorm Asthma in Israel**
Yoav Yair[1],*, Yifat Yair[2], Baruch Rubin[2], Ronit Confino-Cohen[3], Yosef Rosman[3]
Eduardo Shachar[4] and Menachem Rottem[5]
1 – Interdisciplinary Center (IDC) Herzliya, School of Sustainability, Israel
2 – Hebrew University of Jerusalem, Faculty of Agriculture, food and Environment, Rehovot, Israel
3 – Meir Medical Center, Kfar-Saba, Israel
4 - Rambam Medical Center, Haifa, Israel
5 – Ha'Emek Medical Center, Afula, Israel










**\*Corresponding author**
Prof. Yoav Yair
School of Sustainability, Interdisciplinary Center (IDC) Herzliya
P.O. Box 167 Herzliya 4610101 Israel
(p) +972-9-9527952 (m) +972-52-5415091 (f) +972-9-9602401
Email: yoav.yair@idc.ac.il



**Abstract.** We report on the first recorded case of thunderstorm asthma in Israel, that occurred during an exceptionally strong Eastern Mediterranean super-cell thunderstorm on October 25[th] 2015. The storms were accompanied by intensive lightning activity, severe hail, downbursts and strong winds followed by intense rain. The hospital admission records from three hospitals – two in the direct route of the storm (Meir Medical Center in Kfar-Saba and Ha'Emek in Afula) and the other just west of its ground track (Rambam Medical Center in Haifa) showed that the amount of admissions of patients with respiratory problems in the hours immediately following the storm increased compared with the average numbers in the days before. Following the passage of the gust front and the ensuing increase in particle concentrations, within several hours there was a noticeable increase in the number of patients with respiratory problems, in line with the pattern reported by Thien et al., (2018) for the massive epidemic in Perth, Australia. This increase in patient presentation to the ER persisted for 48-72 hours before going back to normal values, indicating that the event was related to the super-cell outflow. We discuss how the likelihood of incidence of such public-health events associated with thunderstorms will be affected by global trends of population growth, urbanization and climate change.

## 1. Introduction.

Thunderstorms and lightning are natural hazards, lethal and destructive with important implications on human societies. They are often accompanied by severe weather, hail and flash flooding that entail significant economic losses (Yair, 2018). Public health effects of thunderstorms that are not related to direct strikes of people are caused by downdrafts during the mature and decay stages of thundercloud evolution. The strong down-winds from the thundercloud, often accompanied by precipitation particles, reach the surface and cause cold outflows. These winds have the potential to eject large concentration of pollen and dust particles into the air, releasing allergens in the size range < 2.5 micrometers. Such particles can be inhaled into the respiratory system and cause an acute allergic response. If occurring during the flowering season of specific plants, this may result in "*Thunderstorm Asthma*" epidemics (Bellomo et al. ,1992; Packe and Ayers, 1995; Venables et al., 1997; Wardman et al., 2002; Dales et al., 2003; D'Amato et al., 2016; 2017), which are expressed as severe respiratory problems, especially in sensitive populations (infants, senior citizens and people with prior allergic susceptibility). During the development stage, updrafts carry surface


aerosols and pollen particles into the cloud, where the high humidity causes them to
rupture. At the mature stage of the thunderstorm, downdrafts and precipitation carry
these fragments to the ground. When the winds impinge on the surface they diverge and
the outflow can enhance the concentrations of airborne particles (when occurring in dry
desert areas this leads to the formation of well-known dust-wall known as "Haboob").
If the storm occurs during flowering season, the gust front below the cloud may release
more pollen from grasses and plants, and then updrafts may entrain them into the cloud
base. Strong electric fields develop in the thunderstorm which can further accelerate
pollen rupture, increasing the risk of exposure to allergens.
Grass pollen is a well-known cause of hay-fever and allergic asthma, and has been
implicated as the cause of two cases of thunderstorm asthma epidemics, in Melbourne
(1987/1989) and in London (1994). However, Suphioglu et al. (1998) stated that grass
pollen is too large to penetrate into the lower airways and trigger the allergic response.
The electric fields and humidity can rupture the pollen particles, releasing 700
fragments that contain the major allergen Lol p 5; They showed a 50-fold increase in
the concentration of starch granules in the atmosphere following rain. They also showed
that free grass allergen molecules interact with ambient pollution particles (diesel
exhaust carbon) offering an additional mode of transport and penetration into human
lower airways.
**Figure 1:** A schematic
description of the mechanism that
enhances the concentrations of
airborne aerosols (either pollution
particles or pollen) ahead of a
mature thunderstorm (Taylor and
Jonsson, 2004).

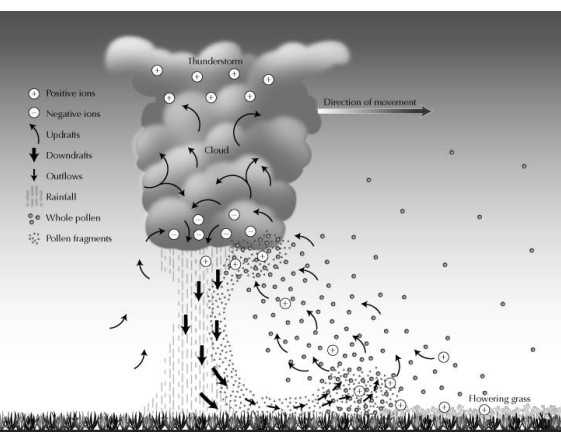

Nasser and Pulimood (2009) reviewed the role of fungal spores such as *Alternaria*
in outbreaks of thunderstorm asthma and showed that the sudden increase in spore
concentrations in the air following large-scale thunderstorm cold flows affects atopic,



sensitized people, and may lead to asthmatic response. There are numerous reports from
many countries about cases of thunderstorm asthma (Dabrera et al., 2012; Andrew et
al., 2017; Beggs, 2017). For example, the Waga-Waga epidemic in Australia on
October 30th 1997 led to 215 ER visits by asthmatic subjects with 41 hospitalizations,
a fact that created an unusual burden on the health services there (Girgis et al., 2000).
The most extreme case on record occurred in Melbourne, Australia, in November 2016
(Thien et al., 2018), when a thunderstorm asthma epidemic following a gust front
induced by thunderstorms resulted in more than 8000 people being admitted to hospitals
for allergy and respiratory problems, with 10 fatalities. Though not directly caused by
lightning as an electrical phenomenon, the allergic response of the population followed
(or was prompted) by a chain-reaction commencing with the dynamics of the cold
outflow from the thunderstorm. D'Amato et al. (2015) characterized the main aspects
of thunderstorm-associated asthma epidemics (based on their Table 2): (a) The
epidemics are limited to seasons when there are high concentrations of airborne
allergenic pollens (b) There is a close temporal association between the start of the
thunderstorm and the onset of the epidemics. (c) There are not high levels of pollution
related gasses and particles during the thunderstorm asthma outbreak (d) People who
stay indoors with windows closed are not affected and (e) there is a major risk for
subjects who are not optimally treated for asthma; subjects with pollen-induced allergic
rhinitis and without prior asthma are also at risk.
While this definition focuses on the allergic responses to airborne pollen or fungal
spores, some reports consider other environmental factors such as humidity,
temperature and pressure changes (Rossi et al., 1993; Ito et al., 1989). Another chemical
effect of lightning activity that may also play a role in thunderstorm asthma epidemics
is the production of significant amounts of NO and $O_3$ near the surface. Lightning-
produced NOx (LtNOx) is an important agent in tropospheric chemistry and is also a
precursor for the production of greenhouse gasses (Price et al., 1997; Boersma et al.,
2005; Ott et al., 2010). Ozone by itself is a potent oxidizer and is known to create severe
respiratory response when inhaled (Molfino et al., 1991; Gleason et al., 2014). Although
it is short-lived and quickly recombines with molecular oxygen, ozone is present near
the surface for several hours after electrical activity, and together with airborne pollen
or pollution particles can induce a synergistic effect on human health. Campbell-
Hewson et al. (1994) considered several types of pollen and fungal spores, but also
ozone concentrations and lightning, in the context of a thunderstorm asthma epidemic



in Cambridge and Peterborough in southern England in June 1994. They reported an
increase by a factor ~2 of ozone concentration (45 ppb compared with daily average of
28.7 ppb) and high pollen counts before the rain and concluded that the causes of the
epidemic were likely multifactorial. It should be pointed out that although there were
37 lightning strikes in that region, the authors did not attribute the rise in ozone
concentrations to lightning but rather to pollution. A thorough review published by the
World Allergy Organization (D'Amato et al., 2015) surveyed the expected changes in
the occurrence of thunderstorm asthma and concluded that people with hypersensitivity
to pollen allergy should be advised to stay indoors when there are clear indications that
thunderstorm activity is expected. Such early-warning capabilities for lightning are
becoming operational in some countries (for example the Lightning Potential Index
[LPI] which is used in WRF; Lynn and Yair, 2010; Lynn et al., 2012), but there seems
to be a gap between forecasting lightning and administrating public-health warnings,
and sensitive populations are not always effectively alerted.
**2.  Data Sources**
We used data from various sources for studying possible correlations between
meteorological conditions, lightning occurrence, aerosol concentrations, pollen counts
and respiratory illnesses.
a.  Lightning data was obtained from the Israeli Lightning Detection Network
(ILDN) operated by the Israeli Electrical Corporation (IEC). The system and its
capabilities are described by Shalev et al. (2011).
b.  Meteorological data – temperature, humidity, wind and pressure data was
obtained from the Israeli Meteorological Service (IMS) for selected stations
throughout the country.
c.  Aerosol data – we used the PM2.5 and PM10 data that are collected routinely
by the Ministry of Environmental Defense in Israel, that operates a national
network of > 40 stations. These stations report particle concentrations at 5-
minute intervals. That system also records Ground Level Ozone data.
d.  Pollen data – The daily average pollen and spore concentrations (number/m$^3$)
were obtained from the Ted Arison Laboratory for Monitoring Airborne
Allergens at Tel-Aviv University. The species are listed in Appendix 1.
e.  Hospital admission records for respiratory symptoms were collected for a
specific list of allergy-related illnesses that can be attributed to airborne particles



in thunderstorm events. The long- term averages were obtained from hospital
records to establish the baseline.

**3. Meteorological Conditions**
The synoptic condition leading to the unusual event described here are summarized
by Razy et al. (2018) and will be briefly described below. During October 24[th] 2015 the
eastern Mediterranean was dominated by a Red-Sea Trough (RST, Ben-Ami et al.,
2014), a low-pressure region extending from the south along the Red-Sea northward to
the eastern Mediterranean. This system transports tropical air toward the Levant region
in the lower-levels. At the Upper-levels, a pronounced trough was situated west of the
Levant. This trough had two effects: One is a transport of tropical air by the south-
southwesterly winds aloft and second is upward motion at the mid-levels, induced by
positive vorticity advection ahead of this trough. Prior to the beginning of the storm, a
cold front was noted west of the Israeli coast. At the same time a meso-scale cyclone
was formed over the Sinai Peninsula and the southeastern Mediterranean. During the
morning hours of October 25[th,] the cyclone, together with the cold front, moved toward
inland. Around 07 UTC this cold front crossed central Israel, accompanied by extremely
developed thunderclouds, with tops reaching 17 km height. The highly populated area
of central Israel, extending from the coastal region inland, was subjected to torrential
rains for 1-2 hours and large hailstorm with over 5cm diameter. The intensity of the
storm can be attributed, at least partly, to the tropical nature of the warm air transported
from south by the RST, ahead of the storm. The super-cell subsided upon reaching the
Jordan rift in eastern Israel. The entire event caused 1 fatality, extensive flooding in
several Israeli cities and agricultural damages. It also impacted the national electrical
grid with power outages lasting up to 3 days in central Israel.
a. **Wind** – Based on the Israeli Meteorological Service data, the storm was typified
by destructive south-westerly winds that exceeded 25 m s$^{-1}$, with gusts of >36
m s$^{-1}$, which can be attributed to the downbursts from the active cells. Figure 2
presents wind speeds measured at several locations. The distance from Tel-Aviv
coast (purple line) to Hadera port (red line) is approximately 40 km, indicating
a very wide gust front that swept across central Israel together with the
movement of the active cells. The sustained high winds lasted for more than two
hours, and caused a significant increase in amounts of airborne particulate
matter (see below).





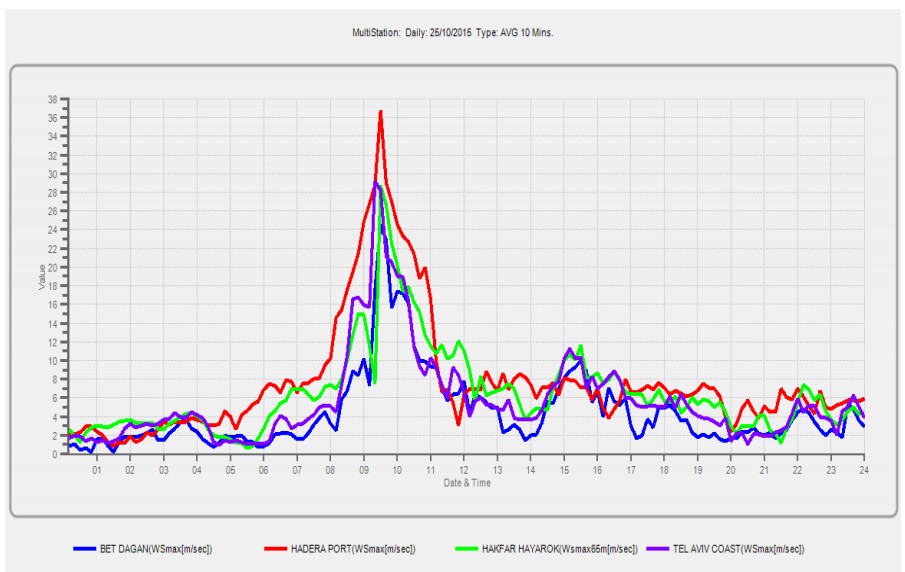


**Figure 2:** Wind speed at 4 different stations along Israel. Bet Dagan (in blue) is located 12 km southeast of Tel-Aviv. Hadera Port (red) is located on the coastline, 45 km north of Tel-Aviv. Hakfar Hayarok (green) is 5 km northeast of Tel-Aviv, and Tel-Aviv coast (purple) is located on the Mediterranean coastline. All stations recorded an abrupt and short-lived increase in wind-speed around 10 AM local time, indicating the passage of the gust front. Data courtesy the Israeli Meteorological Service.

**b. Electrical Activity -** More than 17,000 cloud-to-ground lightning strokes were registered by the ILDN during this event, exceeding the annual total for the entire country (Figure 3). As Figure 4 shows, at the peak of the event the average cloud-to-ground flash rates between 090-0930 LT were greater than 450 strokes per minute. One should consider that this is only the Cloud-to-Ground (CG) flash rate as the ILDN does not record Intracloud flashes (IC). If we accept the ratio of IC/CG reported by Yair et al. (1998), then the total flash rate would be more than 1000 flashes per minute, exceeding the maximum global record of flash rates found in the Argentina-Paraguay border (Zipser et al., 2006). This was the most powerful thunderstorm ever observed in Israel since lightning detection became operational in 1997.






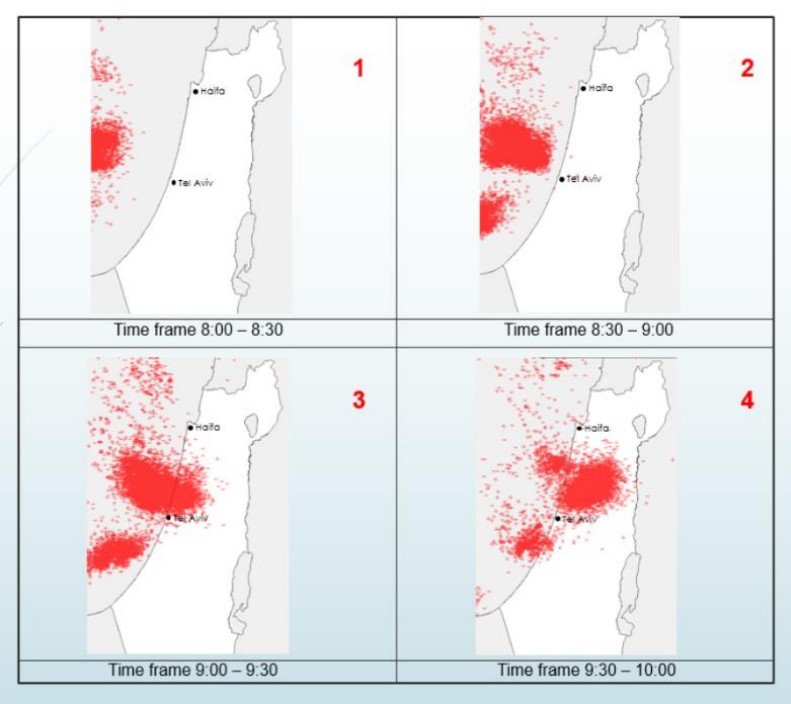


**Figure 3:** Lightning strokes detected on October 25th 2015 by the ILDN (Israel Lightning Detection Network) operated by the Israeli Electrical Corporation. Each point is a ground stroke. The panels show cumulative values at 30 minutes intervals, local time indicated.

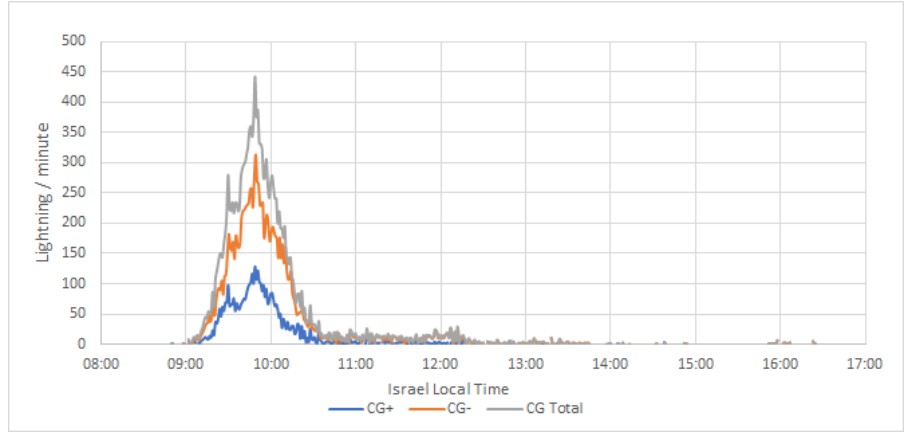


**Figure 4:** 1-minute accumulated lightning numbers detected on October 25th 2015 as a function of local time. The total cloud-to-ground stroke rate (grey) exhibits a sharp maximum around 09:45 local time, as the cells passed over central Israel.





### 4. Particle Concentrations

The results from the Israeli Ministry for Environmental Protection's air-quality
monitoring network show a remarkable increase in the concentrations of PM 2.5
particles, up to 10-fold the normal values (Figure 5). This is due to the very strong winds
ahead of the cells, that picked up considerable amounts of dust, pollen and other types
of aerosols from the surface.

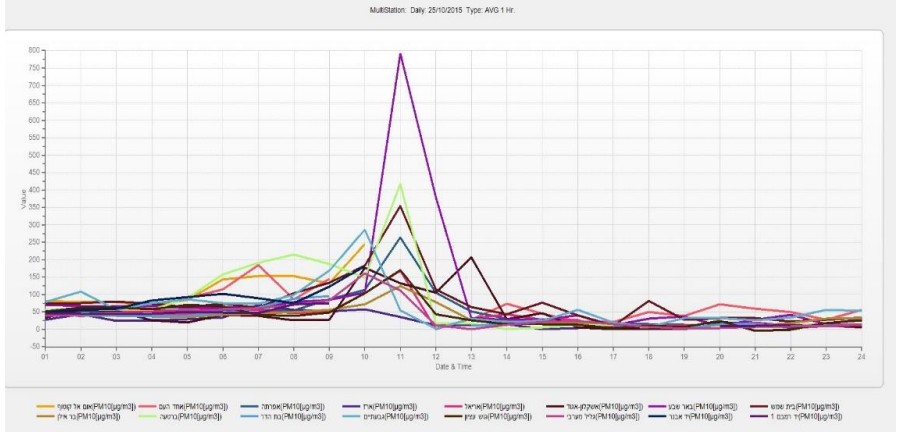


**Figure 5:** Mass concentration of PM10 aerosols for 16 stations in Israel, 25[th] October 2015. Data is given
in µg m[-3]. Note the peak around 1100 local time, coinciding with the passage of the gust front. The sharp,
strong peak was meaured at the Rambam Medical Center in haifa.

The daily pollen amounts for October 2015 (Figure 6) exhibit two significant peaks,
which are related to severe weather events. It should be noted that before the onset of
the storm on October 25[th], there were already larger than usual amounts of pollen and
spores in the air (up be a factor of 3). This supports the thunderstorm asthma hypothesis
of pollen processing inside the storm by humidity and electric fields, that results in
rupture and release of allergens into the cold outflow (D'Amatto et al., 2015; Beggs,
2017). The decrease in pollen concentrations after the storm is explained by washout
and dilution after the rain and winds associated with passage of the active cells. The list
of flowering allergenic plants in October in Israel is presented in Appendix A.



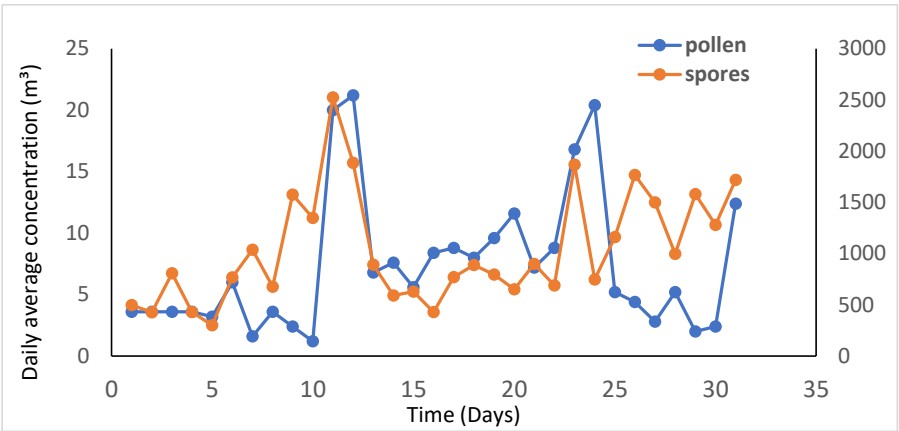


**Figure 6**: Daily average concentrations of pollen and spore numbers for October 2015, based on
data collected at Tel-Aviv University's monitoring station in the botanical gardens on campus (Data
courtesy of Prof. Amram Eshel, the Laboratory for Pollen Monitoring, Tel-Aviv University).

## 271     5. Hospital ER admissions

The hospital admission records of patients with respiratory problems were obtained
from three hospitals. The Meir Medical Center is located in the city of Kfar-Saba
(population 110,000), 15 km north-east of Tel-Aviv in the central coastal plain. The
Ha'Emek Medical Center is located in Afula (population 43,000), a regional urban
center located in an agricultural and rural part of northern Israel, close to Mt. Tabor.
The Rambam Medical Center is located in Israel's largest port city of Haifa (population
280,000) and is the largest of the three. Figure 7 shows the records of a full week with
numbers of patients, starting 3 days before the event. The ER admission records show
that the numbers of presentations of patients on October 25[th] increased compared with
the numbers of the days before the storm. Although in absolute numbers the numbers
may seem low, the values admitted on the day of the thunderstorm represent a clear
deviation from monthly average for October. At the Meir (located just below the
ground-track of the storm cells) and Rambam (located west of the ground-track)
hospitals there was a clear increase in the number of ER presentations which can be
related to the passage of the gust-front in the surrounding areas and the ensuing increase
in particle concentrations. Based on records of arrival times at the ER, we noted that
within several hours after the thunderstorm there was a noticeable increase in the
number of patients with respiratory problems, in line with the pattern reported by
Newson et al. (1997) and Thien et al., (2018). At the Ha'Emek medical center in Afula




there was no significant increase and the numbers were practically the same as the day
before. In all three hospitals, this increase in patient presentation to the ER with
respiratory problems persisted for 24 hours and a clear decline was noticed in the
following day, likely related to a wash-out effect by precipitation that followed the
passage of the active cells. This decline was more pronounced at the Meir and Ha'Emek
hospitals which experienced heavy rains during of the storm, and it lasted for 48 hours.
At the Rambam Medical Center in Haifa the numbers of ER presentations with
respiratory problems rose again to high values, likely to the ambient values of air
pollution related to aerosols in the Bay of Haifa, a well-known source of industrial
emissions (Sa'aroni et al., 2018).

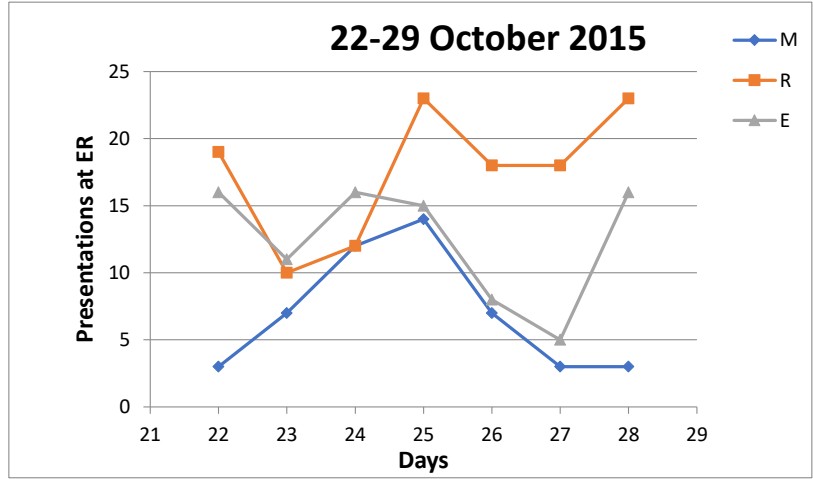

**Figure 7**: Emergency room presentations at 3 Israeli hospitals in the 3 days preceding and following the
October 25th 2015 super-cell event: M = Meir Medical center (blue), R = Rambam medical center
(orange), E = HaEmek medical center (grey).

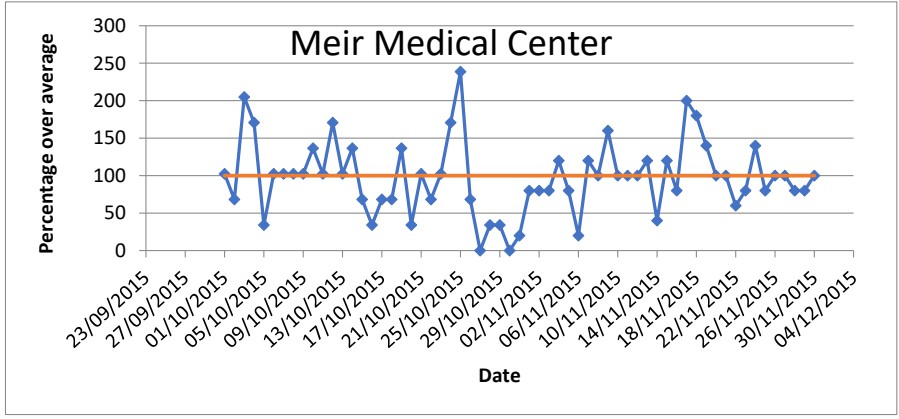



**Figure 8**: Two months of ER presentations of patients with respiratory problems at the Meir Medical
Center in Kfar-Saba, central Israel (for the period 1.10.2015-30.11.2015). The October 25$^{th}$ record shows
a 250% increase above the long term average in a single day.

## 6. Discussion

In most reported cases of thunderstorm asthma in Europe, Canada, US and
Australia, the initiating agents were summer convective storms, and their occurrence
coincided with the flowering season of many plant species whose pollen is known to be
highly allergenic. In Israel, thunderstorms and lightning occurs almost exclusively
during winter months ((December-January-February) and are associated with the
passage of Cyprus Lows or Red-Sea Trough [RST] (Ziv et al., 2008; Shalev et al., 2011;
Yair et al., 2014; Ben-Ami et al., 2015). During these months there is little flowering
and pollen concentrations are low. However, some of the most severe convective events
in Israel occur during fall and spring months, when the RST pressure system transports
mid-level moisture into the eastern Mediterranean and the atmosphere is unstable,
enabling deep convection and intense lightning activity. These events occur mostly in
October-November and March-May, and coincide with flowering of various allergen-
bearing plant species, for example *Ambrosia* spp. (Waisel et al., 1997; Waisel et al.,
2008; Appendix A), and so have the potential to instigate thunderstorm-asthma
epidemics.
The October 25$^{th}$ 2015 super-cell event was by far one of the strongest thunderstorm
episodes ever recorded in Israel. The unique synoptic circumstances of this event
coincided with massive flowering of *Ambrosia* spp. already shown to be highly
allergenic and wide-spread in central Israel (Yair et al., 2017; 2018). Previous studies
showed that the mechanism by which thunderstorm dynamics recycle ambient aerosols
is very effective in releasing allergens from pollen particles, that may otherwise not
reach and affect sensitized populations (Taylor and Jonsson, 2004; D'Ammato et al.,
2015). The strong electric fields that existed during that thunderstorm, manifested by
the high flash rate, likely aided in exploding the outer shell of pollen particles and
enriching the air with allergens, that accompanied other aerosol particles already
present in the environment. The track of the storm passed directly above the densely
populated, mostly urban part of Israel, where the ambient concentrations of pollution
particles was already high. Additionally, as the spore counts indicate (Figure 6), the
background levels of fungal spores, that play an important role in asthma allergenicity
(Packe and Ayers, 1986; Dales et al., 2003), was high the day before the storm. Thus,
it was the convergence of several factors on the particular day that initiated the observed
increase in ER respiratory presentations. Admittedly, the public health data presented
in this study is limited, but follow-up research being presently conducted is bound
enable us to properly identify the characteristics of admitted patients (as performed by
Thien et al., 2018).
What can be done to protect sensitized populations against thunderstorm
asthma, especially in light of the emerging trends of thunderstorm frequency (Romps
et al., 2016; Brooks, 2013; Diffenbaugh et al., 2013; Yair et al., 2018), the extended
period of plant flowering (Ziska et al., 2011) and the increase in allergen content in
pollen (Singer et al., 2005) in a warmer climate? A thorough review published by the
World Allergy Organization (D'Amato et al., 2015) surveyed the expected changes in
the occurrence of thunderstorm asthma and concluded that people with hypersensitivity
to pollen allergy should be advised to stay indoors when there are clear indications that
thunderstorm activity is expected. Silver et al. (2018) examined the seasonality and
predictability of asthma-related admission at Melbourne hospitals, using time-series
ecological approach. They suggest that the observed spring peak in asthma patient
numbers may be related to thunderstorm asthma as they are associated with rainfall,
high humidity, and enhanced grass pollen levels, but the rarity of such events
undermines predictive capabilities. Indeed, early-warning capabilities for lightning are
becoming operational in some countries (for example the Lightning Potential Index
[LPI] which is being used for medium-range weather forecast models; Lynn and Yair,
2010; Lynn et al., 2012) and pollen forecast models are also used to predict the onset
and spread of pollen concentrations (Sofiev et al., 2013; Zhang et al., 2014). However,
there seems to be a gap between a combined forecasting procedure of pollen and
lightning and administrating public-health warnings, and thus sensitive populations
may not be effectively alerted. We therefore suggest to include proper public health
alerts when there is clear indication for the coincidence of thunderstorms during plant
flowering season in specific regions where allergenic species are found.



**Acknowledgment**: This research is supported by the Israeli Science Foundation and
the National Chinese Science Foundation grant 2460/17. Pollen data courtesy Prof.
Amram Eshel, Tel-Aviv University. Lightning data obtained from the Israeli Electrical





Corporation (IEC). We wish to thank Dr. Nurit Keinan for her kind help with Appendix
A.

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



**Figure Captions**

**Figure 1:** A schematic description of the mechanism that enhances the concentrations of airborne aerosols (either pollution particles or pollen) ahead of a mature thunderstorm (Taylor and Jonsson, 2004).

**Figure 2**: Wind speed at 4 different stations along Israel. Bet Dagan (in blue) is located 12 km southeast of Tel-Aviv. Hadera Port (red) is located on the coastline, 45 km north of Tel-Aviv. Hakfar Hayarok (green) is 5 km northeast of Tel-Aviv, and Tel-Aviv coast (purple) is located on the Mediterranean coastline. All stations recorded an abrupt and short-lived increase in wind-speed around 10 AM local time, indicating the passage of the gust front. Data courtesy the Israeli Meteorological Service.

**Figure 3:** Lightning strokes detected on October 25[th] 2015 by the ILDN (Israel Lightning Detection Network) operated by the Israeli Electrical Corporation. Each point is a ground stroke. The panels show cumulative values at 30 minutes intervals, local time indicated.

**Figure 4:** 1-minute accumulated lightning numbers detected on October 25[th] 2015 as a function of time. The total cloud-to-ground stroke rate (grey) exhibits a sharp maximum around 09:45 local time, as the cells passed over central Israel.

**Figure 5:** Mass concentration of PM10 aerosols for 16 stations in Israel, 25[th] October 2015. Data is given in $\mu g\ m^{-3}$. Note the peak around 1000 local time, coinciding with the passage of the gust front. The sharp, strong peak was meaured at the Rambam Medical Center in haifa.

**Figure 6**: Daily average concentrations of pollen and spore numbers for October 2015, based on data collected at Tel-Aviv University's monitoring station in the botanical gardens on campus (Data courtesy of Prof. Amram Eshel, the Laboratory for Pollen Monitoring, Tel-Aviv University).

**Figure 7**: Emergency room presentations at 3 Israeli hospitals in the 3 days preceding and following the October 25[th] 2015 super-cell event: Meir Medical center (blue), Rambam medical center (orange), HaEmek medical center (grey).

**Figure 8**: Two months of ER presentations of patients with respiratory problems at the Meir Medical Center in Kfar-Saba, central Israel (for the period 1.10.2015-30.11.2015). The October 25[th] record shows a 250% increase in a single day.



**Appendix A**

Table showing flowering months for various allergenic plants in Israel (based on
Keinan, 1992). Yellow marks little flowering, dark brown marks massive flowering.

| | 1 | 2 | 3 | 4 | 5 | 6 | 7 | 8 | 9 | 10 | 11 | 12 |
|---|---|---|---|---|---|---|---|---|---|---|---|---|
| *Cynodon dactylon* | Y | Y | O | O | O | O | O | O | O | Y | Y | Y |
| *Hyparrhenia hirta* | | Y | Y | O | O | O | O | O | O | O | Y | Y |
| *Pennisetum clandestinum* | | Y | Y | O | O | O | O | O | O | Y | Y | |
| *Stenotaphrum secundatum* | | | Y | O | O | O | O | O | O | Y | | |
| *Paspalum vaginatum* | | | Y | O | O | O | O | O | O | O | Y | |
| *Zoisia* sp. | | | Y | O | O | O | O | O | O | O | Y | Y |
| *Sorghum halepense* | | | Y | O | O | O | O | O | O | | | |
| *Chloris gayana* | | | Y | O | O | O | O | O | O | | | |
| *Poa* sp. | | | Y | O | Y | | | | | | | |
| *Hordeum* sp. | | | Y | O | Y | | | | | | | |
| *Lolium* sp. | | | Y | O | Y | | | | | | | |
| *Bromus* sp. | | | Y | O | Y | | | | | | | |
| *Dactylis glomerata* | | | Y | Y | Y | | | | | | | |
| *Avena* sp. | | | Y | Y | Y | | | | | | | |
| *Parietaria* sp. | | Y | O | O | O | O | O | O | O | O | Y | Y |
| *Ricinus communis* | | | Y | O | O | O | O | O | O | O | | |
| *Chenopodium* sp. | | | Y | O | O | O | O | O | O | | | |
| *Urtica* sp. | | Y | O | O | O | O | O | O | | | | |
| *Mercurialis annua* | | Y | O | O | O | O | O | O | | | | |
| *Plantago* sp. | | | Y | Y | O | O | O | O | | | | |
| *Amaranthus* sp. | | | | | Y | O | O | O | O | Y | | |
| *Inula viscosa* | | | | | Y | O | O | O | O | O | Y | |
| *Ambrosia* sp. | | | | | Y | O | O | O | O | O | Y | |
| *Xanthium* sp. | | | | | | | Y | O | O | O | | |
| *Salsola kali* | | | | | | | | | O | O | O | |
| *Atriplex halimus* | | | | | | | | | O | O | Y | |
| *Artemisia monosperma* | | | | | | | | | O | O | | |
| *Artemisia herba alba* | | | | | | | | | | Y | O | |
| *Eucalyptus* sp. | Y | Y | Y | Y | Y | Y | Y | Y | Y | Y | Y | Y |
| *Thuja* sp. | Y | O | | | | | | | | | | |
| *Cupressaceae* | Y | O | O | | | | | | | | | |
| *Phoenix dactylifera* | Y | O | O | | | | | | | | | |
| *Quercus ithaburensis* | | | O | O | | | | | | | | |
| *Quercus calliprinos* | | | O | O | Y | | | | | | | |
| *Pistacia lentiscus* | | Y | O | O | | | | | | | | |
| *Pistacia palaestina* | | | O | O | | | | | | | | |
| *Olea europaea* | | | Y | O | O | Y | | | | | | |
| *Acacia* sp. | | | Y | Y | O | | | | | | | |
| *Carya illinoinensis* | | | | Y | Y | | | | | | | |
| *Ailanthus glandulosa* | | | | Y | O | O | | | | | | |
| *Ceratonia siliqua* | | | | Y | Y | | | | | O | O | Y |
| *Schinus* sp. | | | | | | | | | Y | O | Y | |
| *Casuarina sp.* | | | | | | | | | | Y | Y | |