# Peer review of "1. Introduction."

_Natural Hazards and Earth System Sciences, 2019_

## Referee Comment (RC1) · Anonymous Referee #1 · 26 May 2019

**1   Summary and review**

The authors describe what seems to be an episode of thunderstorm asthma, apparently the first such reported case in Israel. The authors have covered most areas that may be expected in such a case study, and such case studies should be published. Thunderstorm asthma is a rare phenomenon, and only by pooling knowledge across the international community can we develop a sufficiently rigorous understanding for prediction of these events. However I am suggesting that it be subject to major revisions, not for any one reason but for a range of moderate and minor issues that probably should be fixed before it be published.

I would also question whether this is the right journal for this particular topic. Natural Hazards and Earth System Science isn't really a health-focussed journal, in my under-

standing. My searches haven't revealed anything on asthma or thunderstorm asthma in particular. This is really up to the editors, but I think it would probably be better placed elsewhere.

**2 Major comments**

- There is an overabundance of review articles on thunderstorm asthma, relative to the number of original case studies. There are quite a lot of theories about the exact causation of this phenomenon. Multiple factors are no doubt at play, but there are difficulties in the scales of what can be observed. The proposed mechanisms span a vast r ange of scales (rupturing of bioaerosols, inhalation of microscopic particles, advection and transport of these particles over potentially considerable distances and large heights), and many of these phenomena are difficult to measure in the laboratory, let alone in the uncontrolled environment. As such, I would recommend that the authors treat the published theories of the causes of thunderstorm asthma as theories that have only limited support, rather than as well established.

- Several of the figures seem of relatively poor quality (especially figures 2, 3 and 5), with poor resolution, small legend text, legend text that isn't self-explanatory and lines that aren't obviously distinguishable when printed in black and white.

- The Introduction talks a fair bit about ozone and NOx, but this doesn't feature in the results section.

- The results section doesn't look at any particular pollen or fungal spore taxa. I would recommend providing more information about what pollen types were present on these days. It might be worth showing counts if they are available. If not, I would suggest recounting these slides if they are still available. Pollens

Interactive
comment

vary in their allergenicity and other attributes, and it would be of interest to know if the thunderstorm asthma reported was likely linked to some particular taxa. Your Appendix A provides some limited insights, but I would like to see more.

- The results for the aerosol concentrations appear to show $PM_{10}$ concentrations (Figure 5), rather than $PM_{2.5}$ concentrations, as mentioned in the text (L247). An increase in coarse particles (as would be seen in a "raised dust incident" during a severe gust front) does not necessarily lead to a big jump in $PM_{2.5}$ levels. It may be worth presenting results for both the coarse and the fine fractions.

- Were the hospital data available for longer periods? If so, one can do some statistical analysis to detect spikes in asthma-related presentations, which may help identify other (previously unreported) epidemic asthma episodes.

- Were hospital data available for sites that were not directly in the storm path? If so, it would be informative to compare what was experienced at these locations.

Minor comments:

- Consider adding a map (in the appendix, at least) showing the locations of all the sites referred to in the text.

- L43-46: there is some repetition here, probably best avoided in the Abstract.

- L47: Melbourne, not Perth.

- L48: Don't introduce an acronym in the abstract when it won't be used again in the abstract. Don't use an acronym before introducing it, even if it's an obvious one in your field.

- L50-52: I think this is a bit of over-reach. The article doesn't really do this, and even if it did, it would be quite speculative.

[Figure]

- L58: "Public health heffects of thunderstorms not related to direct strikes of people are caused by downdrafts..." – what about flash flooding, damaging winds, large hail and landslides? These can cause some fairly serious harm to people?

- L58-59: Somewhere around here I think it should be mentioned that thunderstorm asthma is both rare and quite limited in its scale compared to the burden of asthma more generally.

- L62: "eject" → "result in the release of"

- L69-78: This section needs many more references.

- L77-78: please cite the article supporting this claim.

- L81: The Suphiglou (1998) reference (note: no co-authors, so you can delete the "et al") was in large part copied (directly rather than paraphrased) from Knox (1993; Clinical and Experimental Allergy, Volume 23, pages 354-359). I would suggest at least citing Knox (1993) as well. Given the egregious nature of this case of plagarism, the authors may choose to the Knox (1993) paper instead.

- L83-84: references to support this statement are lacking.

- L110: "Waga-Waga" → "Wagga Wagga"

- L111: It may be worth qualifying the number of presentations and admissions by also stating the population of Wagga Wagga at the time (which should be available from the Australian Bureau of Statistics).

- L115: The number '8000 people being admitted'. Please double-check your sources. I suspect that this may be the total number of admissions or presentations for all causes (not just asthma-related). A more relevant figure may be the percentage increase in asthma-related admissions or presentations associated

with this event, or the excess number of admissions or presentations associated with the event.

- L122: "allergenic pollens" → "allergenic pollens and/or fungal spores"

- L134: "... and is also a precursor for the production of greenhouse gases". First, this is tangential to the topic. Second, it's unclear which greenhouse gases you are talking about. Third, from what is written it is unclear whether this is a major contributor to any of the major greenhouse gases.

- L147-148: "the authors ... to pollution". Ozone production requires sunlight, hydrocarbons and NOx. Lightning provides only the NOx. Anthropogenic pollution isn't the only source of volatile organic compounds, but is indeed responsible for the more intense concentrations. The authors should review this article to see if their critique is warranted.

- L153: "WRF" – see my comment above about acronyms.

- L158-160: Please specify what time period and geographic regions was studied. Were all the data available for the full period? Were some data available for longer periods?

- L174-177: How many hospitals? What time-period? Was this just the total admissions or specific ICD codes? If the latter, which ICD codes? Did the authors need or obtain ethics approval for this research? If not, please note. If so, please cite the ethics authority.

- L184: "Levant region". Please consider marking this on the map suggested above.

- L185: "lower levels", "upper levels": please quantify this statement. What pressure levels or heights above ground level do you mean?

- L194-195: "torrential rains": it would be worth stating the rain rate or rainfall total over this period.

- L227-228: I would suggest moving this sentence earlier in the paragraph, such as after the first sentence. It provides additional perspective.

- Figure 4: For this journal, doese it really matter how many positive and negative lightning strikes were recorded. Would the total strike count not suffice?

- L257-265 and Figure 6: See my comment above about being more specific about which pollen and fungal taxa are present.

- L272-300: Please be more consistent (or at least clearer) about whether these were 'admissions' or 'presentations'. The hospital records may not differentiate this. As I understand it, a 'presentation' occurs when somebody arriving and asking for treatment, whereas an 'admission' occurs when somebody has seen the triage nurse and then been treated by a doctor.

- L272-300: I'm unclear whether the results reported are for all diagnoses, or only those related to asthma and allergic respiratory diseases. See my comment above about ICD codes.

- L288-289: See the previous comment. Without knowing if these are all diagnoses or asthma-related, this statement isn't fully justified.

- L296: "and it lasted" → "lasting"

- L298: "likely to" → "likely due to"

- L298-299: "air pollution related to aerosols". The Introduction talks about the influence of ozone and NOx, but these concentrations aren't reported in this paper. Do you think that ozone and NOx played a role here? If not, I'm not sure that its prominence in the Introduction is warranted.

- L313: "summer" → "spring or summer"

- L318-319: "During these months there is little flowering and pollen concentrations are low." – References would help.

- L331: "showed → "suggest". See my first comment under 'Major comments'

- L333: "D'Ammato" → "D'Amato"

- L335-337: "in exploding the outer shell of pollen particles and enriching the air with allergens, that accompanied other aerosol particles already in the environment → "in rupturing the pollen membranes and enriching the air with respirable allergens".

- L340: "that play an important role in asthma allergenicity" → "that may play an important role in triggering allergenic asthma"

- L344-345: "is bound enable us to properly identify" → "will help us to understand"

- L362: "the Lightning Potential Index [LPI] which is being used for medium-range..." → "the Lightning Potential Index, as calculated by some numerical..."

- Please write PM10 and PM2.5 as $PM_{10}$ $PM_{2.5}$

---

## Author Comment (AC1) · 20 Aug 2019

Response to Reviewer #1

We wish to thank the reviewer for his/her insightful comments and suggestions. We accepted most of the listed comments and revised the manuscript accordingly. Our detailed responses and changes are listed below, and are marked in the new version.

General Comment The authors describe what seems to be an episode of thunderstorm asthma, apparently the first such reported case in Israel. The authors have covered most areas that may be expected in such a case study, and such case studies should be published. Thunderstorm asthma is a rare phenomenon, and only by pooling knowledge across the international community can we develop a sufficiently rigorous understanding for prediction of these events. However, I am suggesting that it be subject to major revisions, not for any one reason but for a range of moderate and minor issues

that probably should be fixed before it be published. I would also question whether this is the right I would also question whether this is the right journal for this particular topic. Natural Hazards and Earth System Science isn't really a health-focused journal, in my understanding. My searches haven't revealed anything on asthma or thunderstorm asthma in particular. This is really up to the editors, but I think it would probably be better placed elsewhere.

Answer: Thunderstorm asthma epidemics constitute a natural hazard, and may be regarded as rare – perhaps as rare as strong volcanic eruptions or tsunamis. Nevertheless, it is clearly a public hazard that is related to thunderstorms, which are frequent and prone to increase in a warmer climate (see Yair, ERL, 2018). There is a robust knowledge-base on the biological-medical causes of thunderstorm asthma, as evident by the rapidly growing number of papers on this topic. Likely thunderstorm asthma is under-reported, because individual cases that do not accumulate to an epidemic-scale hospital admissions event are obscured by the day-to-day variability. However, this is not the focus of the present paper, which highlights the geophysical circumstances leading to this unique public health event in Israel. I would like to argue – and therefore disagree with this reviewer - that the NHESS is the proper journal for publishing our results, because of its interdisciplinary nature and its accessibility to wide audiences of different disciplines. Adding this dimension to the scope of topics regarded as "natural hazards" by introducing a human dimension is, in my mind, an asset, not a deficiency. This is of course an editorial decision and I believe that by sending the paper out for review had already been taken. The table below indicates the growing interest in this topic in the last couple of years, a matter that illustrates the interest in the scientific community in it.

number of articles related to thunderstorm asthma years

3 1990-1994 11 1995-1999 45 2000-2004 112 2005-2009 201 2010-2014 413 2015-2019

[Figure]

Major comments 1. There is an overabundance of review articles on thunderstorm asthma, relative to the number of original case studies. There are quite a lot of theories about the exact causation of this phenomenon. Multiple factors are no doubt at play, but there are difficulties in the scales of what can be observed. The proposed mechanisms span a vast range of scales (rupturing of bioaerosols, inhalation of microscopic particles, advection and transport of these particles over potentially considerable distances and large heights), and many of these phenomena are difficult to measure in the laboratory, let alone in the uncontrolled environment. As such, I would recommend that the authors treat the published theories of the causes of thunderstorm asthma as theories that have only limited support, rather than as well established.

Answer: We agree with the reviewer that the precise mechanism by which allergens are released from plant pollen and affect humans is still under investigation. Moreover, it could well be that several different processes occur simultaneously in different circumstances. Laboratory experiments on the effects of electric fields and humidity on pollen rupture are certainly needed, and such were actually included in a proposal we submitted to the Israeli Science Foundation (regrettably, not funded). In order to account for the uncertainty of the exact cause of thunderstorm asthma, we rephrased several of the paragraphs describing the hypothetical mechanisms.

2. Several of the figures seem of relatively poor quality (especially figures 2, 3 and 5), with poor resolution, small legend text, legend text that isn't self-explanatory and lines that aren't obviously distinguishable when printed in black and white.

Answer: These Figures were redone and their quality improved.

3. The Introduction talks a fair bit about ozone and NOx, but this doesn't feature in the results section.

Answer: The fact that ozone production accompanies lightning activity was reviewed in the Introduction so as to exemplify the complex nature of the topic. As in the previous comment by the reviewer, we do not subscribe to any single mechanism that may be at

work in thunderstorm asthma, but rather wish to present the readers with multiple facets of the phenomenon. Attribution and causality are especially hard in clinical studies, and the present research does not attempt to address this aspect. For this reason, we choose to leave the text on ozone production here, even if it was not evaluated in the present research and so not mentioned in the Results.

4. The results section doesn't look at any particular pollen or fungal spore taxa. I would recommend providing more information about what pollen types were present on these days. It might be worth showing counts if they are available. If not, I would suggest recounting these slides if they are still available. Pollens vary in their allergenicity and other attributes, and it would be of interest to know if the thunderstorm asthma reported was likely linked to some particular taxa. Your Appendix A provides some limited insights, but I would like to see more.

Answer: Yes, we have the entire data set of pollen and fungal spores (30 species), identified and attributed to the different taxa (Ambrosia, Alternaria, Ascospore…etc.). This specific dataset can be shared with the reviewer upon request, but we do not think that it is needed in the manuscript. See an excerpt here:

Date Ambrosia Arecaceae Artemisia Asteraceae Casuarina Chenopo-Amaran Cupressaceae 2015-10-01 0 0 0.4 0 0 0 0.4 2015-10-02 0 0.4 0.8 0 0 0.8 0 2015-10-03 0.4 0 0.4 0 0 0 0 2015-10-04 0 0 0 0 0 0 0.4 2015-10-05 0 0 0 0 0 0.8 0 2015-10-06 0 0.4 0.4 0 0 0.4 0 2015-10-07 0 0 0.4 0 0 0 0

5. The results for the aerosol concentrations appear to show PM10 concentrations (Figure 5), rather than PM2:5 concentrations, as mentioned in the text (L247). An increase in coarse particles (as would be seen in a "raised dust incident" during a severe gust front) does not necessarily lead to a big jump in PM2:5 levels. It may be worth presenting results for both the coarse and the fine fractions.

Answer: We obtained the PM2.5 data for numerous ground stations for the same period of time (e.g. 25.10.2015), and the results clearly show that there was a parallel rise in

fine particle concentrations during the same time that of the peak in the PM10 mass loading shown in Figure 5. The results are now presented in Figure 5b and addressed in the text.

6. Were the hospital data available for longer periods? If so, one can do some statistical analysis to detect spikes in asthma-related presentations, which may help identify other (previously unreported) epidemic asthma episodes.

Answer: Yes, the data is now available and it took us months to retrieve it. We attach below a month-long hospital admission records from the 3 medical centers we worked with. There is at least one other conspicuous daily maximum in the number of patients with allergies, but that event was unrelated to thunderstorms or any other outstanding meteorological event. Much longer periods are extremely hard to obtain for the present scope of this study, but will be pursued in a wider survey of thunderstorm asthma in Israel.

7. Were hospital data available for sites that were not directly in the storm path? If so, it would be informative to compare what was experienced at these locations.

Answer: No, we did not request or receive other hospital's data. However, inspired by the reviewer's comment, we will apply and try to obtain such information from other hospitals, more remote from the storm's direct path. This may take months and will not be included in this manuscript.

Minor Comments. Consider adding a map (in the appendix, at least) showing the locations of all the sites referred to in the text. Answer: we re-did Figure 3 such that the locations of the hospitals are marked.

L43-46: there is some repetition here, probably best avoided in the Abstract. Answer: thank you for alerting us on the repetition. It is now amended and the abstract was rearranged.

L47: Melbourne, not Perth. Answer: Corrected.

L48: Don't introduce an acronym in the abstract when it won't be used again in the abstract. Don't use an acronym before introducing it, even if it's an obvious one in your field. Answer: Corrected. The text now spells ER as "emergency room".

L50-52: I think this is a bit of over-reach. The article doesn't really do this, and even if it did, it would be quite speculative. Answer: Agree. We changed to: We discuss how the likelihood of incidence of such public-health events associated with thunderstorms will be affected by global trends of in lightning occurrence.

L58: "Public health effects of thunderstorms not related to direct strikes of people are caused by downdrafts..." – what about flash flooding, damaging winds, large hail and landslides? These can cause some fairly serious harm to people? Answer: We rephrased this sentence, such that it now reads: "Public health effects of thunderstorms that are not related to direct strikes of people (or that are caused by the accompanying phenomena mentioned above) may be the result downdrafts during the mature and decay stages of thundercloud evolution..."

L58-59: Somewhere around here I think it should be mentioned that thunderstorm asthma is both rare and quite limited in its scale compared to the burden of asthma more generally.

L62: "eject" -> "result in the release of" Answer: Modified as suggested.

L69-78: This section needs many more references. Answer: We added several references and the text is rephrased accordingly.

L77-78: please cite the article supporting this claim. Answer: there are several papers on the rupture mechanism, we refer to Knox (1993), Taylor et al., (2004a) and Miguel et al. (2006).

L81: The Suphiglou (1998) reference (note: no co-authors, so you can delete the "et al") was in large part copied (directly rather than paraphrased) from Knox (1993; Clinical and Experimental Allergy, Volume 23, pages 354-359). I would suggest at least

citing Knox (1993) as well. Given the egregious nature of this case of plagiarism, the authors may choose to the Knox (1993) paper instead. Answer: We thank the reviewer for alerting us to this case of plagiarism, of which we were totally unaware. The text was changed, the Suphiglou (1998) reference deleted and we refer to the Knox (1993) paper as recommended.

L83-84: references to support this statement are lacking. Answer: Added

L110: "Waga-Waga" -> "Wagga Wagga" Answer: Corrected

L111: It may be worth qualifying the number of presentations and admissions by also stating the population of Wagga Wagga at the time (which should be available from the Australian Bureau of Statistics). Answer: We actually do not see value of this information here, and furthermore we have no access to data older than 2004 through the ABS website.

L115: The number '8000 people being admitted'. Please double-check your sources. I suspect that this may be the total number of admissions or presentations for all causes (not just asthma-related). A more relevant figure may be the percentage increase in asthma-related admissions or presentations associated with this event, or the excess number of admissions or presentations associated with the event.

Answer: Thank you for finding this typo: the number of ED (Emergency Department) presentations should have been >3000 and not as written. The actual number was 3365, as described in Thien et al. (2018) and in Table 1 of Harun et al. (2019).

L122: "allergenic pollens" -> "allergenic pollens and/or fungal spores" Answer: Corrected

L134: "... and is also a precursor for the production of greenhouse gases". First, this is tangential to the topic. Second, it's unclear which greenhouse gases you are talking about. Third, from what is written it is unclear whether this is a major contributor to any of the major greenhouse gases. Answer: Correct, this is indeed tangential. That part

of the sentence was omitted.

L147-148: "the authors ... to pollution". Ozone production requires sunlight, hydrocarbons and NOx. Lightning provides only the NOx. Anthropogenic pollution isn't the only source of volatile organic compounds, but is indeed responsible for the more intense concentrations. The authors should review this article to see if their critique is warranted. Answer: Although lightning produces tropospheric ozone and it may also play a role in the complex interactions leading to thunderstorm asthma epidemics, we follow the reviewer's suggestion and diminish the text on this aspect (relevant references deleted). Still we choose to mention this aspect and hence mention the Hewson et al. paper.

L153: "WRF" – see my comment above about acronyms. Answer: WRF is the well-known initial of the Weather and Forecasting Research model. This is now explicitly written in the sentence.

L158-160: Please specify what time period and geographic regions was studied. Were all the data available for the full period? Were some data available for longer periods?

L174-177: How many hospitals? What time-period? Was this just the total admissions or specific ICD codes? If the latter, which ICD codes? Did the authors need or obtain ethics approval for this research? If not, please note. If so, please cite the ethics authority. Answer: the text was rephrased and contains the requested details: "Hospital presentation records for patients with respiratory symptoms of specific ICD codes at the Emergency Room (ER) were collected from 3 Israeli hospitals for a specific list of allergy-related illnesses. Approvals of the internal Helsinki committee in each hospital were obtained. The long- term averages were obtained from hospital records to establish the baseline."; [The list of diagnoses is attached at the end of this response letter.]

L184: "Levant region". Please consider marking this on the map suggested above. Answer: We changed the wording of this sentence: "This system transported tropical

air toward Egypt, Jordan, Israel, Lebanon and Cyprus in the lower-levels (850 hPa)."

L185: "lower levels", "upper levels": please quantify this statement. What pressure levels or heights above ground level do you mean? Answer: Done. We refer to 850 hPa and 500 hPA, respectively.

L194-195: "torrential rains": it would be worth stating the rain rate or rainfall total over this period. Answer: The following sentence was added: Rain-gauge data obtained from the Israeli Meteorological Service show that in several places in central Israel the 10-minute rain rate exceeded 100 mm h-1 with a total of >50 mm in the entire event (constituting ~10% of the annual average).

L227-228: I would suggest moving this sentence earlier in the paragraph, such as after the first sentence. It provides additional perspective. Answer: Done as recommended.

Figure 4: For this journal, does it really matter how many positive and negative lightning strikes were recorded. Would the total strike count not suffice? Answer: We agree that in the context of this manuscript the polarity has no importance, and the graph was changed accordingly.

L257-265 and Figure 6: See my comment above about being more specific about which pollen and fungal taxa are present. Answer: This is information is now included in the Appendix.

L272-300: Please be more consistent (or at least clearer) about whether these were 'admissions' or 'presentations'. The hospital records may not differentiate this. As I understand it, a 'presentation' occurs when somebody arriving and asking for treatment, whereas an 'admission' occurs when somebody has seen the triage nurse and then been treated by a doctor. Answer: Thank you for this comment, which we accepted and acted upon. All the medical data was of patient presentations at the ER, and not actual admittance.

L272-300: I'm unclear whether the results reported are for all diagnoses, or only those

related to asthma and allergic respiratory diseases. See my comment above about ICD codes. Answer: We had a well-defined list of diagnoses, agreed upon by all co-authors (three of which are the allergy department heads in their respective hospitals), and this is attached as an Appendix at the end of this response letter.

L288-289: See the previous comment. Without knowing if these are all diagnoses or asthma-related, this statement isn't fully justified. Answer: True, but as written in the response to the previous comment, there is a specific list of respiratory-system related diagnoses. We slightly edited this sentence.

L296: "and it lasted" -> "lasting" Answer: Done.

L298: "likely to" -> "likely due to" Answer: Done.

L298-299: "air pollution related to aerosols". The Introduction talks about the influence of ozone and NOx, but these concentrations aren't reported in this paper. Do you think that ozone and NOx played a role here? If not, I'm not sure that its prominence in the Introduction is warranted. Answer: Correct, we do not claim that ozone and NOx played a role. This was also addressed in our response to a previous comment, and the manuscript was changed so as to diminish the volume dedicated to LtNOx and ozone.

L313: "summer" -> "spring or summer" Answer: Done.

L318-319: "During these months there is little flowering and pollen concentrations are low." – References would help. Answer: Done. We refer to Keinan (1992).

L331: "showed -> "suggest". See my first comment under 'Major comments' Answer: Done.

L333: "D'Ammato" -> "D'Amato" Answer: Done.

L335-337: "in exploding the outer shell of pollen particles and enriching the air with allergens, that accompanied other aerosol particles already in the environment -> "in

rupturing the pollen membranes and enriching the air with respirable allergens". Answer: Done.

L340: "that play an important role in asthma allergenicity" -> "that may play an important role in triggering allergenic asthma" Answer: Done.

L344-345: "is bound enable us to properly identify" -> "will help us to understand" Answer: Done.

L362: "the Lightning Potential Index [LPI] which is being used for medium range..." -> "the Lightning Potential Index, as calculated by some numerical..." Answer: Done.

Please write PM10 and PM2.5 as PM10 PM2:5 Answer: Done.

Appendix A – Types of medical conditions associated with particle pollution (Starting with: acute bronchiolitis)

Please also note the supplement to this comment:
https://www.nat-hazards-earth-syst-sci-discuss.net/nhess-2019-137/nhess-2019-137-AC1-supplement.pdf

---

## Referee Comment (RC2) · Anonymous Referee #2 · 30 Aug 2019

**Review of the paper "First reported case of Thunderstorm Asthma in Israel" by Yoav Yair, Yifat Yair, Baruch Rubin, Ronit Confino-Cohen, Yosef Rosman, Eduardo Shachar, and Menachem Rottem, submitted to NHESS.**

The paper refers to the first recorded case of thunderstorm asthma in Israel. It presents new interdisciplinary research findings that can contribute a lot in the knowledge of health problems associated to meteorological factors. The analysis of thunderstorms integrating lightning activity is not new; some literature already shows the relationship between some thunderstorms and respiratory problems; but the combination of all of them in order to relate the cycle of life of this convective system and health crisis evolution is new. Besides this, the paper discusses how the likelihood of incidence of such public-health events associated with thunderstorms will be affected by global trends of population growth, urbanization and climate change. The paper is well supported by recent literature, the scientific methodology followed and the different expertise of the authors.

Authors say that the event occurred during an exceptionally strong super-cell thunderstorm accompanied by intensive lightning activity, severe hail, downbursts and strong winds followed by intense rain. They have found that the amount of admissions of patients with respiratory problems in the hours immediately following the passage of the gust front. They also discuss how the likelihood of incidence of such public-health events associated with thunderstorms will be affected by global trends of population growth, urbanization and climate change.

Attending that the other referee has focused their comments on aspects related with the medical and health aspects and has provided a detailed list of comments and suggestions, I will focus my review on meteorological aspects.

**Abstract**

Write the meaning of ER

**Introduction**

Page 2, line 3. Add "human losses" ("that entail significant economic and human losses"). There is a great number of references about fatalities due to this kind of events (i.e. Petrucci, O., L.Aceto, C.Bianchi, V.Bigot, R. Brázdil, S.Pereira, A.Kahraman, Ö.Kılıç, V.Kotroni, M.C. Llasat, M. Llasat-Botija, K. Papagiannaki, A. A. Pasqua, J. Řehoř, J. Rossello Geli, P. Salvati, F. Vinet, J.L. Zêzere, 2019. Flood Fatalities in Europe, 1980–2018: Variability, Features, and Lessons to Learn, Water 2019, 11, 1682; doi:10.3390/w11081682)

Page 2, lines 4-5. There are numerous public health effects of thunderstorms, because they can be related with heavy rainfalls and floods, hailstorms or tornadoes. In those cases, health effects can be drowning and heart attacks -see the previous suggested reference-, impact of direct strikes, GOLPES due to the collapse of trees or walls, or by objects transported by the wind, direct impact of severe hail, car accidents, and so on. Consequently, you should modify this sentence, perhaps including some literature references to other health effects of thunderstorms.

Page 2, lines 4-5. I would recommend a little modification of the Introduction. You start the physical explanation that relates the cycle of life of the thunderstorm with the asthma, speaking about "downdrafts during the mature and decay stages of thundercloud evolution". In line 15

you introduce the "development stage" (that is anterior to mature and decay stages) and afterwards again the downdrafts. Taking into account that this paper is addressed to scientist from different disciplines, and some of them probably doesn't know the cycle of life of thunderstorms, it can create some confusion. I would suggest moving line 15 to a new paragraph focused on the short explanation of this cycle of life and its relation with the causes of asthma (pollen, etc)

Page 3, line 5. Add a reference on "the formation of well-known dust-wall known as "Haboob"". It is not so well-known by all the potential readers of this paper.

Page 4, line 7. Remind the interdisciplinary and multidisciplinary character of this paper. Some people cannot know was it a "gust front". Explain here or in the paragraph in which you explain the phases of the thunderstorm. Taking into account that this event affected more than 8000 people it should be a big gust front, probably due to a multicellular thunderstorm, a supercellular thunderstorm or a mesoscale system. Then, it would be better to say "induced by a thunderstorms system"

Page 4, lines 9-12. Why do you speak here about lightning activity?

Page 4, line 23. You say "Another chemical effect of lightning activity…" but I don't find where you have introduced any chemical effect of lightning activity. Attending your expertise, I consider that it would be interesting to write a paragraph explaining the chemical effects of lightning activity that can be related with asthma.

Page 5, line 6. As you are starting with recommendations and warning systems, I would recommend un PUNTO Y APARTE before "A thorough review published…"

Page 5, line 12. Probably some readers didn't know what WRF is. It would be better to write "which is used in the meteorological model WRF to forecast thunderstorm activity" or something like this.

Meteorological Conditions

Page 6, line 9. Write "upper levels" (they are not only the level of 500 hPa, usually they arrive until 300 hPa)

Page 6. What was the role of the mesoscale cyclone? Probably the organization of the flow that helped the advection of wet air form the Sea. Which factor triggered the convection? The cold front? Please, clarify. Are you sure that it was a supercell and not a multicellular system or mesoscale system? Usually supercells are related with severe weather (i.e. tornadoes) and have a mesocyclonic circulation inside. Could you check it? It would be interesting to include a satellite image showing the thunderstorm and the micro-front or squall line created by the downbursts

Page 7. You say that it "was the most powerful thunderstorm ever observed in Israel since lightning detection became operational in 1997". This fact merits to be included in the abstract and conclusions.

Page 7. You explain here the role of humidity and electric fields, and the fact that after the thunderstorm that results in rupture and release of allergens into the cold outflow. I think that

it would be useful to comment this in the Introduction linking the cycle of life of the thunderstorm with the evolution of formation and dispersion of pollen and the other pollutants.

**Discussion**

Page 12. You say that in Israel, thunderstorms and lightning occurs almost exclusively during winter months but afterwards you say that some of the most severe convective events in Israel occur during fall and spring months, and that in both cases they are associated to the RST pressure system. I would suggest to substitute "exclusively" by "mainly" and checking if RST is present in the three seasons.

---

## Author Comment (AC2) · 15 Sep 2019

Response to reviewer #2

We wish to thank the reviewer for his/her insightful comments and suggestions. We accepted most of the listed comments and revised the manuscript accordingly. Our detailed responses and changes are listed below, and are marked in the new version.

Abstract Write the meaning of ER Answer: Done (*emergency room)

Introduction Page 2, line 3. Add "human losses" ("that entail significant economic and human losses"). There is a great number of references about fatalities due to this kind of events (i.e. Petrucci, O., L.Aceto, C.Bianchi, V.Bigot, R. Brázdil, S.Pereira, A.Kahraman, Ö.KÄślÄśç, V.Kotroni, M.C. Llasat, M. Llasat-Botija, K. Papagiannaki, A. A. Pasqua, J. ÅŸehoř, J. Rossello Geli, P. Salvati, F. Vinet, J.L. Zêzere, 2019. Flood

[Figure]

Fatalities in Europe, 1980–2018: Variability, Features, and Lessons to Learn, Water 2019, 11, 1682; doi:10.3390/w11081682) Answer: Added, as suggested.

Page 2, lines 4-5. There are numerous public health effects of thunderstorms, because they can be related with heavy rainfalls and floods, hailstorms or tornadoes. In those cases, health effects can be drowning and heart attacks -see the previous suggested reference-, impact of direct strikes, hits due to the collapse of trees or walls, or by objects transported by the wind, direct impact of severe hail, car accidents, and so on. Consequently, you should modify this sentence, perhaps including some literature references to other health effects of thunderstorms. Answer: We accept this suggestion and made a thorough re-write of the Introduction. Relevant references were added.

Page 2, lines 4-5. I would recommend a little modification of the Introduction. You start the physical explanation that relates the cycle of life of the thunderstorm with the asthma, speaking about "downdrafts during the mature and decay stages of thunder-cloud evolution". In line 15 you introduce the "development stage" (that is anterior to mature and decay stages) and afterwards again the downdrafts. Taking into account that this paper is addressed to scientist from different disciplines, and some of them probably doesn't know the cycle of life of thunderstorms, it can create some confusion. I would suggest moving line 15 to a new paragraph focused on the short explanation of this cycle of life and its relation with the causes of asthma (pollen, etc). Answer: Same as above, we re-wrote the entire Introduction.

Page 3, line 5. Add a reference on "the formation of well-known dust-wall known as "Haboob"". It is not so well-known by all the potential readers of this paper. Answer: Added (Williams et al. 2007), and the text was re-worded.

Page 4, line 7. Remind the interdisciplinary and multidisciplinary character of this paper. Some people cannot know was it a "gust front". Explain here or in the paragraph in which you explain the phases of the thunderstorm. Taking into account that this event affected more than 8000 people it should be a big gust front, probably due to a multicellular thunderstorm, a supercellular thunderstorm or a mesoscale system. Then, it would be better to say "induced by a thunderstorms system" Answer: The text was corrected as suggested.

Page 4, lines 9-12. Why do you speak here about lightning activity? Answer: Good comment, the text was changed and the mention of lightning omitted.

Page 4, line 23. You say "Another chemical effect of lightning activity…" but I don't find where you have introduced any chemical effect of lightning activity. Attending your expertise, I consider that it would be interesting to write a paragraph explaining the chemical effects of lightning activity that can be related with asthma. Answer: The text was revised, and in line with reviewer #1 comments, the reference to surface chemical effects of lightning was diminished. We did not study any effects of NO and O3 in the present research, so it was not justified to discuss them in depth. We added the following sentence: " This aspect of lightning activity was not considered in the present study."

Page 5, line 6. As you are starting with recommendations and warning systems, I would recommend un PUNTO Y APARTE before "A thorough review published…" Answer: Done.

Page 5, line 12. Probably some readers didn't know what WRF is. It would be better to write "which is used in the meteorological model WRF to forecast thunderstorm activity" or something like this. Answer: Done.

Meteorological Conditions Page 6, line 9. Write "upper levels" (they are not only the level of 500 hPa, usually they arrive until 300 hPa) Answer: The text was modified: "This system transported tropical air toward Egypt, Jordan, Israel, Lebanon and Cyprus in the lower-levels (850 hPa). At the upper-levels (500 hPa), a pronounced trough was situated with the axis slanted between Crete and Cyprus"

Page 6. What was the role of the mesoscale cyclone? Probably the organization of

the flow that helped the advection of wet air form the Sea. Which factor triggered the convection? The cold front? Please, clarify. Are you sure that it was a supercell and not a multicellular system or mesoscale system? Usually supercells are related with severe weather (i.e. tornadoes) and have a mesocyclonic circulation inside. Could you check it? It would be interesting to include a satellite image showing the thunderstorm and the micro-front or squall line created by the downbursts. Answer: We added a satellite image obtained by the MODIS instrument on-board the TERRA satellite (Figure 3a) showing the multicell system above Israel. We also modified the text to better explain the role of the mesocyclone.

Page 7. You say that it "was the most powerful thunderstorm ever observed in Israel since lightning detection became operational in 1997". This fact merits to be included in the abstract and conclusions. Answer: Done, a sentence was added to the abstract.

Page 7. You explain here the role of humidity and electric fields, and the fact that after the thunderstorm that results in rupture and release of allergens into the cold outflow. I think that it would be useful to comment this in the Introduction linking the cycle of life of the thunderstorm with the evolution of formation and dispersion of pollen and the other pollutants. Answer: Done as suggested. In light of earlier comments and those of Reviewer #1, the Introduction was completely re-written.

Discussion Page 12. You say that in Israel, thunderstorms and lightning occurs almost exclusively during winter months but afterwards you say that some of the most severe convective events in Israel occur during fall and spring months, and that in both cases they are associated to the RST pressure system. I would suggest to substitute "exclusively" by "mainly" and checking if RST is present in the three seasons. Answer: Done as suggested.

———————————————————

**Fig. 1.** Figure 3a

---

## Author Comment (AC3) · 15 Sep 2019

[revised manuscript text omitted]

---

## Editor Decision (ED1)

[revised manuscript text omitted]